# A Translational Approach to Spinal Neurofibromatosis: Clinical and Molecular Insights from a Wide Italian Cohort

**DOI:** 10.3390/cancers15010059

**Published:** 2022-12-22

**Authors:** Rosina Paterra, Paola Bettinaglio, Arianna Borghi, Eleonora Mangano, Viviana Tritto, Claudia Cesaretti, Carla Schettino, Roberta Bordoni, Claudia Santoro, Sabrina Avignone, Marco Moscatelli, Mariarosa Anna Beatrice Melone, Veronica Saletti, Giulio Piluso, Federica Natacci, Paola Riva, Marica Eoli

**Affiliations:** 1Molecular Neuroncology Unit, Fondazione IRCCS Istituto Neurologico Carlo Besta, 20122 Milano, Italy; 2Department of Medical Biotechnology and Translational Medicine (BIOMETRA), University of Milan, Segrate, 20122 Milan, Italy; 3Institute of Biomedical Technologies (ITB) National Research Center (CNR), ITB-CNR, Segrate, 20122 Milan, Italy; 4Medical Genetics Unit, Woman-Child-Newborn Department, Fondazione IRCCS Ca’ Granda-Ospedale Maggiore Policlinico, 20122 Milano, Italy; 5Department of Advanced Medical and Surgical Sciences, 2nd Division of Neurology, Center for Rare Diseases and InterUniversity Center for Research in Neurosciences, University of Campania “Luigi Vanvitelli”, 81100 Naples, Italy; 6Neurofibromatosis Referral Center, Department of Woman, Child and of General and Specialized Surgery, and Clinic of Child and Adolescent Neuropsychiatry, Department of Mental Health, Physical and Preventive Medicine, University of Campania “Luigi Vanvitelli”, 81100 Naples, Italy; 7Neuroradiology Department, Fondazione IRCCS Ca’ Granda-Ospedale Maggiore Policlinico, 20122 Milano, Italy; 8Neuroradiology Department, Fondazione IRCCS Istituto Neurologico Carlo Besta, 20133 Milano, Italy; 9Sbarro Institute for Cancer Research and Molecular Medicine, Center for Biotechnology, Temple University, Philadelphia, PA 19122, USA; 10Developmental Neurology Unit, Mariani Foundation Center for Complex Disabilities, Fondazione IRCCS Istituto Neurologico Carlo Besta, 20133 Milan, Italy; 11Department of Precision Medicine, Università degli Studi della Campania “Luigi Vanvitelli”, 81100 Naples, Italy

**Keywords:** neurofibromatosis type 1, spinal neurofibromatosis, neurofibroma, spinal tumors, *NF1* pathogenic variant

## Abstract

**Simple Summary:**

At present, no systematic study of the clinical spectrum and molecular characteristics of NF1 patients with spinal neurofibromatosis (SNF), a phenotypic subclass of neurofibromatosis 1 (NF1), has been carried out. Here, we provide evidence that SNF patients are at high risk of problematic neurofibromas, presenting not only bilateral neurofibromas involving all spinal roots, but also a higher incidence of internal neurofibromas and nerve root swelling. From a histopathological view, not only neurofibromas, but also neurogangliomas are present in SNF. The analysis of 19 families with at least 1 member affected by SNF showed a high phenotypic variability within the SNF families. Furthermore, we discovered a higher prevalence of missense mutations in SNF compared to classical NF1. Both clinical features and genetic testing can help in identifying cases at risk of SNF, and that are more likely to benefit from a spinal MRI scan.

**Abstract:**

Spinal neurofibromatosis (SNF), a phenotypic subclass of neurofibromatosis 1 (NF1), is characterized by bilateral neurofibromas involving all spinal roots. In order to deepen the understanding of SNF’s clinical and genetic features, we identified 81 patients with SNF, 55 from unrelated families, and 26 belonging to 19 families with at least 1 member affected by SNF, and 106 *NF1* patients aged >30 years without spinal tumors. A comprehensive *NF1* mutation screening was performed using NGS panels, including *NF1* and several RAS pathway genes. The main features of the SNF subjects were a higher number of internal neurofibromas (*p* < 0.001), nerve root swelling (*p* < 0.001), and subcutaneous neurofibromas (*p* = 0.03), while hyperpigmentation signs were significantly less frequent compared with the classical NF1-affected cohorts (*p* = 0.012). Fifteen patients underwent neurosurgical intervention. The histological findings revealed neurofibromas in 13 patients and ganglioneuromas in 2 patients. Phenotypic variability within SNF families was observed. The proportion of missense mutations was higher in the SNF cases than in the classical NF1 group (21.40% vs. 7.5%, *p* = 0.007), conferring an odds ratio (OR) of 3.34 (CI = 1.33–10.78). Two unrelated familial SNF cases harbored in trans double *NF1* mutations that seemed to have a subclinical worsening effect on the clinical phenotype. Our study, with the largest series of SNF patients reported to date, better defines the clinical and genetic features of SNF, which could improve the management and genetic counseling of NF1.

## 1. Introduction

Neurofibromatosis type 1 (NF1; MIM 1622009), one of the most frequent genetic diseases [1,2], is caused by heterozygous mutations of the *NF1* tumor suppressor gene, and is characterized by highly variable expressivity. At present, a clinically significant genotype–phenotype correlation is found in approximately 10% of cases: patients with the NF1 microdeletion syndrome (MIM 613675) [3]; subjects with a deletion of a specific, single amino acid (c2970_2972del p. Met992del) [4,5]; and individuals with missense mutations affecting arginine at position 1809 [6,7,8], codons 844–848 [9], or affecting pMet1149, pArg1276, and p.Lys1423 [10].

In 2014, Ruggieri [11] reviewed all cases published up until the time of writing, and defined the clinical criteria of another subtype of the disease: spinal neurofibromatosis (SNF). It is characterized by bilateral neurofibromas involving all spinal roots and few, if any, cutaneous manifestations. This feature allows one to specifically distinguish SNF from neurofibromatosis type 1 (NF1) and multiple neurofibromas few spinal root (MNFSR), presenting a single or few isolated spinal neurofibromas. SNF entails greater morbidity than the classical NF1.

More recently, in a large descriptive study on spinal lesions in NF1, 37 new SNF cases were reported [12]. A high incidence of paraspinal plexiform tumors was observed in those patients, and 23.8% of them had a history of previous spinal surgery. In fact, spinal tumors can remain asymptomatic for years; however, once severe neurological deficits have developed, the likely success of any surgical treatment is greatly reduced. Nevertheless, the value of performing routine spinal MRI scanning at the time of the diagnosis is still controversial, and within the current guidelines, no specific surveillance is foreseen.

Furthermore, SNF is an example of the high intrafamilial variability observed in NF1: SNF patients can belong to families with classical NF1 or to families with all affected members with SNF. Even if some clinical features might be used to predict internal neurofibroma onset [13], so far, there are no reliable patterns to distinguish patients at risk of developing SNF. An association of SNF with missense or splicing mutations has been reported [11,14,15,16], but further studies need to be carried out to confirm those correlations. In addition, in individuals with missense mutations affecting *NF1* codons 844–848 or pArg1276, multiple spinal neurofibromas were more common [9,10].

The aim of this study was to describe the clinical and genetic features characteristic of sporadic and familial cases of SNF, and to also attempt comparisons between the SNF phenotype and classical NF1 groups.

## 2. Materials and Methods

### 2.1. Individuals and Phenotypic Data

A total of 74 individuals with a diagnosis of NF1, according to the revised diagnostic criteria for neurofibromatosis type 1 [17], and of the SNF phenotype, according to Ruggieri’s criteria, were identified using the IRCSS C. Besta Neurological Institute, the IRCCS Ca ’Granda Foundation Ospedale Maggiore Policlinico, and the Azienda Ospedaliera Universitaria dell’Università degli Studi della Campania “Luigi Vanvitelli” electronic databases. Forty-one individuals had sporadic NF1, while thirty-three had a familial form of the disease. All other affected members that were alive and available were clinically assessed, and a cohort of 27 affected relatives (11 MNFSR, 7 SNF, and 9 classical NF1) were added to the original patient set, making it possible to identify 19 NF1 families with at least 1 member affected by SNF, and a total of 81 SNF subjects. In Appendix A, the family pedigrees are reported. Two families had four affected members, five families had three affected relatives, and twelve families had two.

To analyze the phenotype and genotype association, an additional cohort of all NF1 patients present in the database without spinal tumors and aged >30 years (106 cases) was included in the study. An age cut-off value of 30 years was chosen because the probability that they could develop an SNF was very low [13]. In addition, we compared the phenotypes of the SNF cases with the previously described large-scale NF1-affected individuals with “classical” NF1 [1,18,19,20,21,22,23,24,25,26,27,28,29,30,31] already used in other genotype–phenotype studies on NF1 [9,10].

All medical records were surveyed. The data were collected at the time of the genetic screening and reverified for accuracy at the time of this study. We focused on symptoms and signs possibly related to spinal neurofibromas, such as pain that could be attributed to spinal abnormalities or tumors; neurological symptoms, including weakness, sensory deficit, and changes in tone and reflexes; in addition to the already well-known NF1 characteristics: CAL; skinfold freckling; cutaneous, subcutaneous, and plexiform neurofibromas; Lisch-nodules; visual impairment; epilepsy; cognitive impairment; optic nerve glioma (OPG); other neoplasms of the central nervous system and other organs; and skeletal and vascular abnormalities. In particular, a neurofibroma is defined as plexiform when it grows along the length of a nerve, and may involve multiple fascicles and branches. The date of birth, gender, age at the time of the last visit, and the mode of inheritance were also recorded. All patients underwent brain and spinal MRI scans with gadolinium.

Cases with missing data for a particular sign and/or symptom were classified as “unknown” and, consequently, excluded from that part of the genotype–phenotype analysis.

This study was approved by the Fondazione IRCCS Istituto Neurologico Carlo Besta Ethical Committee and Scientific Board (N°50-19/3/2018).

### 2.2. Prediction of Effect of Mutations

The effects on genes and proteins of the mutations identified were predicted based on the Mutation Taster (http://www.mutationtaster.org) (accessed on 13 April 2021) and HGMD (Human Genome Mutation Database—Institute of Medical Genetics, Cardiff, Wales, UK; http://www.hgmd.cf.ac.uk/ac/index.php (accessed on 4 October 2022)), and LOVD (Leiden Open Variation Database; http://www.LOVD.nl.NF1) (accessed on 4 October 2022) [32,33] databases were interrogated to verify whether the mutations were novel.

Novel variants with amino acid changes were further examined for their disease-causing potential using PolyPhen-2 (https://genetics.bwh.harvard.edu./pph2/;) (accessed on 13 October 2022) [34]. The possible effects on mRNA (canonical and noncanonical splicing mutations) were evaluated with splice site prediction conducted by a neural network (http://www.fruitfly.org/seq_tools/splice.html) (accessed on 13 October 2022) [35], and with the Human Splicing Finder (HSF; http://www.umd.be/HSF/) (accessed on 13 October 2022) [36] and the ESE Finder (http://rulai.cshl.edu/cgibin/tools/ESE3/esefinder.cgi?process=home) (accessed on 13 October 2022) [37] tools.

We assessed the clinical significance of the sequence variants according to the American College of Medical Genetics (ACMG)/Association of Molecular Pathology (AMP) guidelines [38].

To investigate the significance of variants classified as “uncertain” in the Clinvar database (https://www.ncbi.nlm.nih.gov/clinvar) (accessed on 13 October 2022), we performed a segregation analysis of the familial *NF1* mutations and predicted the functional consequence of missense variants using ANNOVAR (v.2019Oct24) [39], which included prediction scores from 20 prediction algorithms (SIFT, SIFT4G, Polyphen2-HDIV, Polyphen2-HVAR, LRT, MutationTaster2, MutationAssessor, FATHMM, MetaSVM, MetaLR, PROVEAN, FATHMM-MKL coding, FATHMM-XF coding, fitCons, M-CAP, PrimateAI, DEOGEN2, BayesDel no AF, BayesDel add AF, ClinPred, and LIST-S2). In order to assess the frequencies of the variants in the genera l population, the GnomAD v.2.1.1 (https://gnomad.broadinstitute.org) and 1000 genomes (https://www.internationalgenome.org, release 2015aug) databases (accessed on 17 February 2021) were used.

### 2.3. Statistical Analysis

The X^2^ test and two-tailed Fisher’s exact probability test were used to compare categorical variables, with a *p*-value of < 0.05 considered as statistically significant. The odds ratios (ORs) and their 95% confidence intervals (CIs) were calculated. The genotype–phenotype associations were studied using multiple logistic regression. The Benjamini–Hochberg (B_H) method with false discovery rates of 0.25, 0.05, and 0.01 was used to correct *p*-values for multiple testing. The descriptive, frequency, and comparative statistical analyses were carried out using SPSS 22.0 software.

## 3. Results

### 3.1. Demographics

On 30 June 2020, 768 subjects affected by NF1 and followed by the IRCSS C. Besta Neurological Institute, the IRCCS Ca ’Granda Foundation Ospedale Maggiore Policlinico, and the Azienda Ospedaliera Universitaria dell’Università degli Studi della Campania “Luigi Vanvitelli” underwent a spinal MRI scan. In 220 (28.6%) cases, the MRI was reviewed by a specialist neuroradiologist and showed spinal neurofibromas; in 81 (36.8%) cases, bilateral neurofibromas involving all spinal roots were present (SNF) (Figure 1); and in 139 (63.2%) cases, a single or a few isolated spinal neurofibromas were detected.

### 3.2. Clinical Characteristics of the SNF Cohort

The demographic and clinical characteristics were analyzed in all 81 SNF (26 female and 55 male) patients, 55 from unrelated families and 26 belonging to 19 families with at least 1 member affected by SNF. The median age was 35, ranging from 15 to 74 years. The SNF patients’ clinical features are reported in Table 1.

The spinal neurofibromas were symptomatic in 44 out of 81 (54.3%) cases. In most cases, internal neurofibromas (45.7%) and nerve root swelling (32.1%) were also found. SNF patients underwent the spinal MRI scan at a median age of 28 (3–57) years, and the median follow-up duration was 98 (6–341) months. In total, 15 out of 81 (18.5%) SNF patients underwent spinal surgery (10 at the cervical level, 2 at the cervical and lumbar levels, and 3 at the lumbar level) at a median age of 25 (12–45) years, after a median time of 2 (1–86) months from the first spinal MRI scan. The reason for surgery was not only the tumor growth according to REINS criteria [40], but also the report of myelopathy. The histopathological diagnosis was neurofibromas in 13 cases and ganglioneurofibromas in 2 cases.

A total of 15/81 (18.5%) cases had less than 6 CALS, and 50/81 (47.3%) had no freckling; 9/81 (11.1%) had neither more than 5 CALS nor freckling, but they fulfilled the revised diagnostic criteria for neurofibromatosis type 1 [17] due to the presence of other clinical signs, such as neurofibromas and Lisch nodules. Only 17/81 (21%) individuals had more than 10 CALS.

Cutaneous NFs were present in 88.9% of patients, and, in most cases, in low numbers: less than 11 NFs in 36/81 (44.4%) patients. Subcutaneous neurofibromas were observed in 71.6% of individuals. Both cutaneous and subcutaneous NFs were present in 53/81 (65.4%) cases. Plexiform neurofibromas were observed in 41/81 (50.6%) cases.

For 74 patients, data on the presence or absence of Lisch nodules were available: they were present in 59/74 (79.7%) cases.

Symptomatic and asymptomatic OPGs were observed in 2.5% (2/81) and 6.2% (5/81) of subjects, respectively. Gliomas other than OPGs were present in 9/81 (11.1 %) of patients (4 brainstem gliomas, 1 glioblastoma, 3 pilocytic astrocytomas, and 1 subependimal astrocytoma); 2 cases had both optic and brainstem gliomas. Other tumors different from central nervous system tumors were observed in 12 individuals (pheocromocytomas in 3, MPNST in 3, pancreatic endocrine cancer in 2, and pleomorphic liposarcoma in 1 case).

The most common skeletal abnormality was scoliosis, present in 34/81 (42%) cases; 2 patients had bone dysplasia, and the other 2 exhibited dural ectasia (2.5%). Hydrocephalus was found in 6/81 (7.4%) cases, syringomyelia in 1 case, and Arnold Chiari in another (1.2%). Neuropathy was present in 9 out of 80 (11.3%) cases, headache was reported in 8/80 (10%) cases, and 4 patients had epilepsy (4.9%). Facial dysmorphic features were observed in 8/81 (9.9%) cases, macrocephaly in 19/81 (23.5%), short stature in 6/81 (7.4%), and overgrowth in 1/81 (1.2%).

Neurodevelopmental delay was reported in 9/80 patients (11.2%), and learning disability in 11/78 (14.1%).

Hypertension was the most common cardiovascular disorder, observed in 7/81 (8.6%) patients; heart and vessel malformation were found in 3/78 (3.8%) and 10/75 (13.3%) patients, respectively.

### 3.3. Comparisons of the Clinical Characteristics Observed in the SNF Cohort, the Cohort of the “Classical” NF1 Phenotype from Our Institutions, and in Previously Described Classical NF1 Cohorts from the Literature

The clinical characteristics of our SNF cohort were compared with those observed in a cohort of classical NF1 patients (i.e., patients without spinal tumors) followed by the same institutions. There were 68 females and 38 males with a median age of 47 (31–75) years. Furthermore, when data were available, the clinical features were also correlated with those previously reported in large-scale NF1 classical cohorts that had already been used in other genotype–phenotype studies [1,18,19,20,21,22,23,24,25,26,27,28,29,30,31]. All comparisons are reported in Table 2.

The main features of our SNF cases were a higher number of internal neurofibromas (45.7 vs. 6.6%; *p* < 0.001) as well as nerve root swelling (32.1 vs. 2.8; *p* < 0.001) compared to our classical NF1 cohort. In our classical NF1 cohort, patients with spinal NF were deliberately excluded; therefore, no symptoms such as low back pain or neurological deficit related to spinal involvement were reported. However, the frequency of symptomatic spinal neurofibromas reported in two previous studies in which patients with spinal tumors were included [18,24] were significantly lower (54.3% vs. 1.7 and 1.8; *p* < 0.001). In addition, within skeletal abnormalities in our cohort, scoliosis was more frequent, while other abnormalities were more rare than previously reported in large-scale cohorts of classical NF1 [18,23,24,29].

As concerns pigmentary manifestations, in our SNF cases, the numbers of patients with freckling (61.7% vs. 81.1% and 84.2%; *p* = 0.003 and *p* < 0.001) and with >5 CALs (81.5% vs. 93.4% vs. 89%; *p* = 0.012 and *p* = 0.013) were significantly less than those already reported [4] or observed by us in classical NF1. Conversely, plexiform neurofibromas (50.6% vs. 38.7% vs. 18.6%; *p* < 0.001) and subcutaneous neurofibromas (71.6% vs. 56.6% vs. 57.7%; *p* = 0.035 and *p* = 0.018) were more frequently observed in SNF patients, even if the comparisons, after B_H correction, remained statistically significant only against the previously reported cohort [18,24,28,29].

Cognitive impairment and/or learning disabilities were statistically lower in the SNF cases than in the classical NF1 population reported by previous papers (10% vs. 44.8%; *p* < 0.001) [18,24], but were not different from that observed in our classical NF1 cohort (12.9%).

The SNF patients, as well as the cases with classical NF1 followed by our institutions, had a higher risk of being affected by tumors different from OPG and by cardiovascular abnormalities compared to the classical cohorts reported in the literature [24,26]. These findings probably reflect the differences in the case selection and not specific phenotype features.

### 3.4. Phenotypic Variability within SNF Families

We identified 19 NF1 families with at least 1 member affected by SNF. In Appendix A, the family pedigrees are reported. Two families had four affected members, five families had three affected relatives, and twelve families had two. Overall, 26 patients had SNF, 12 had MNSFR, and 9 had a classical form of the disease.

We observed a phenotypic variability within the SNF families. In most families (11/19), all NF1 individuals were affected by SNF or MNFSR.

As proposed by Ruggieri, we called families “pure SNF families” when all affected members had SNF, “partial SNF families” when all affected members had SNF or MNSFR, and “multiple phenotype families” when at least one member had SNF and the other affected members had MNSFR or classical NF1.

Two families (No. 2 and 12) were pure SNF families. We identified nine partial SNF families (No. 1, 4, 6, 7, 10, 13, 15, 16, and 19), and eight families were multiple phenotype families (No. 3, 5, 8, 9, 11, 14, 17, and 18).

No phenotype differences were observed between SNF cases belonging to “pure”, “partial”, or “multiple phenotype” families, or between SNF patients included in the 19 families and the others SNF patients.

### 3.5. Mutation Analysis

Mutational analysis for *NF1* was performed in 208 NF1 cases (81 SNF, 11 MNFSR, and 115 classical NF1). *NF1* variants were observed in 204 cases. In four other cases, in all those affected by SNF and fulfilling the diagnostic criteria for NF1 [17], no *NF1* causative variants were identified. We identified 160 different *NF1* variants. The *NF1* variants observed are reported in Table 3, Table 4 and Table 5 along with the molecular details (DNA, RNA, and protein change) and the classification of the variants by type, tertile [41], and domain [42,43,44]. In Table 3, 19 families with at least 1 member affected by SNF are reported; Table 4 presents 55 SNF cases; and in Table 5 106 classical NF cases are described.

Seventeen causative variants detected in the classical patients and twenty-eight in the SNF (thirteen belonging to the SNF families) were never reported; the others were already described (Table 3, Table 4 and Table 5).

Four causative variants were present in both unrelated SNF and classical patients: c.1318C > T (425, 356); c.2033_2034dupC (2146, 663, 1238); c.5546G > A (1957, 1065, 1660, 213); and c.6789_6792delTTAC (1877, 1327, 183).

Interestingly, five patients showed more than one *NF1* variant. For three familial cases belonging to three families, it was possible to infer whether one or both *NF1* alleles were affected. Family 1, family 17, and family 18 were informative in answering the above question.

Precisely, in family 1, the SNF patient 1136 (proband), who carried the c.62T > A (pLeu21His) *NF1* missense variant inherited from his MNFSR-affected father (1139) and shared by his SNF-affected brother (1140), showed a second c.528T > A (p.Asp176Glu) *NF1* missense mutation inherited from his mother, indicating that the two missense variants were in trans. Despite his mother (1141) not being affected by NF1, according to the revised diagnostic criteria for neurofibromatosis type 1 [17], the p.Asp176Glu substitution was predicted to be damaging by 9/20 predictors, and may have a subclinical significance. Accordingly, patient 1136 showed a more severe phenotype than his affected father and brother.

In family 17, the SNF patient N04 (proband) presented c.3314 + 2T > C splicing *NF1* variants inherited from her MNFSR-affected mother N05, and showed a second c.7532C > T (p.Ala2511Val) *NF1* missense variant, predicted to be damaging by 10/20 predictors inherited from her father (never clinically evaluated) and shared by her brother N06, displaying a cutaneous NF1 form. The other brother N03, harboring only the c.3314 + 2T > C splicing NF1 variants inherited from the mother, was also affected by SNF, as the proband, but the clinical phenotype was less severe: no internal or plexiform neurofibromas were present. In addition, in this case, the second mutation could have a subclinical effect that could worsen the clinical phenotype of the proband carrying mutations on both *NF1* alleles.

In family 18, the SNF patient N07 (proband) showed the pathogenic c.1595T > G (p.Leu532Arg) and the uncertain c.3242C > G (p.Ala1081Gly) *NF1* missense variants, both shared by his sister, patient N08, and inherited by his nephew, patient N09, indicating that the two variants were in cis. The evidence that both the sister and her child were affected by classical NF1 suggests that this double-mutated allele is not specifically associated with a specific NF1 form.

Two sporadic SNF patients were carriers of two concomitant variants in the *NF1* gene, but we were not able to define their phase as in cis or in trans, because their parents were not available for a segregation study. Precisely, the patient 2207 (2207) presented a pathogenic stop variant and the missense variant *NF1* c.1246 C > T (p.Arg4016*) and c.403C > T (p.Arg135Trp). Even if the second variant was reported as “uncertain” in Clinvar, this variant has been classified as potentially damaging by 19 predictors out of the 20 interrogated, and it is absent from controls in the GnomAD and in the 1000 genomes (1000g2015aug_eur) databases. Moreover, the variant replaces the conserved basic amino acid arginine at residue 135 to polar-neutral tryptophan. The patient 891 (891) had a pathogenic frameshift *NF1* variant c.6346_6347insA (p.Ser2116Tyr*6) and the missense variant *NF1* c.5221G > A (p.Val1741Ile), classified as potentially damaging by 8 out of 20 predictors questioned and not reported either in the GnomAD or in the 1000 genomes databases. The variant replaces the conserved hydrophobic and aliphatic amino acid valine with hydrophobic and aliphatic isoleucine at residue 1741 in the PH domain of NF1.

### 3.6. Comparison of the NF1 Variants between the SNF and the Classical NF1 Cohorts

We chose one case for each family, the proband, and we compared the *NF1* causative mutations, excluding uncertain variants, recorded in the SNF and classical NF1 patients. The numbers of SNF patients with large deletion (microdeletion type 1 and atypical microdeletion), frameshift, missense, nonsense, splicing, and small deletion/insertion mutations were 7 (10%), 17 (24.3%), 15 (21.4%), 10 (14.2%), 19 (27%), and 2 (2.9%), respectively. The proportion of missense pathogenic variants was higher in the SNF cohort than in the classical NF1 group (*p* = 0.007; OR 3.34; CI 1.33–8.38), while the proportion of nonsense mutations was lower (*p* = 0.055; OR 0.46; CI 0.20–1.03). Furthermore, after applying the Benjamini–Hochberg correction for multiple testing with a false discovery rate of 0.05, the first differences remained statistically significant (Table 6).

Furthermore, we added our analysis the data concerning the distribution of *NF1* causative variants in 49 SNF patients already published in the literature and reported by Ruggieri [11]. Among the 49 patients, we chose 1 case (the proband) for each family and all of the reported sporadic patients for a total of 25 SNF patients. The combined analysis with our data (Table 7) showed a statistically significant increase in missense mutations (25.3% vs. 7.5%; *p* = 0.001; OR 4.14; CI 1.76–9.75) in the SNF cohort compared to our classical patient cohort; the *p*-value remains statistically significant after correcting with the Benjamini–Hochberg correction method for multiple testing with a false discovery rate at 0.025 and 0.01.

The proportion of truncating variants was lower (*p* = 0.03; OR 0.44; CI 0.21–0.93) and the proportion of nontruncating variants was higher (*p* = 0.036; OR 2.24; CI 1.04–4.81) in the SNF patients when compared to those observed in the classical NF1 cases (Table 8).

We also investigated whether the risk of developing SNF was associated with causative variants in one particular *NF1* domain (Table 9).

The patients with harboring variants in the HLR domain had a mild to higher risk of developing SNF (*p* = 0.025; OR 2.5; CI 1.1–5.64) (Table 9). Furthermore, after placing the HLR domain and the HLR-CTD domain together, the risk of developing SNF correlated with variants in those domains (34.3% vs. 16%; *p* = 0.006; OR 2.74; CI 1.31–5.72). The difference remained significant after Benjamini–Hochberg correction with an FDR of 0.25. As concerns the causative variant types within the nineteen SNF families, of the six families with at least two SNF cases, in two families, missense variants were observed; in another two, exon deletions were observed; in one, a splicing mutation was observed; and in another one, a frameshift mutation was observed. Conversely, in the six multiple phenotype families, five out of six mutations were truncating.

## 4. Discussion

To our knowledge, this is the largest series of SNF patients reported to date. All cases were followed by three different Italian institutions conforming to the same clinical protocol. In fact, at present, no systematic study of the clinical spectrum and molecular characteristics of NF1 patients with the SNF phenotype has been carried out, even though spinal neurofibromatosis was first described by Pulst in 1991 [45] and by Poyhonen in 1997 [46]. In 2015, Ruggeri established precise criteria to identify this distinct phenotype, and enucleated 49 cases with a true SNF phenotype out of 98 spinal NF cases already described in the literature [11]. Recently, Curtis-Lopez described 37 cases that met the criteria for SNF in a retrospective review of 303 NF1 patients with different types of spinal lesions; unfortunately, no data on NF1 gene mutations were reported [12].

The reported frequency of spinal neurofibromas in NF1 ranges from 15.9% [47] to 65% [25]. In our series, the prevalence of spinal neurofibromas of any type (i.e., single, few isolated neurofibromas, and SNF) and the prevalence of the SNF phenotype alone was 28.6% and 10.5%, respectively. The prevalence of the SNF phenotype was similar to those reported by Curtis-Lopez (12.2%) [12], confirming that, if rigorous diagnostic criteria are applied, SNF is a rare distinct phenotype of NF1. On the contrary, the frequency of spinal neurofibromas in our study was lower than those previously reported in [25] and by Curtis-Lopez [12], at 65% and 58.1%, respectively; however, this is in line with Well (39.6%), who performed whole-body MRI scans on the studied patients [31]. This discrepancy may reflect the differences in selecting patients to undergo a spinal MRI scan.

In our cohort, spinal neurofibromas were symptomatic in 54.3% of the cases. A varied prevalence of the symptomatic spinal tumors in NF1 patients has been described in the literature, ranging from to 2% [25] to 24% [12]; however, both patients with SNF and MNFSR were included by the authors. These findings further demonstrate the effect of spinal abnormalities on affected patients, and emphasize the necessity to evaluate routinely performed MRI scans regarding these abnormalities.

Only a minority of those patients undergo surgery: 18.5% in our study and 23.85% in that by Curtis-Lopez [12]. In 15 cases in our cohort, spinal tumors were removed; the diagnosis was neurofibromas in 13 cases and ganglioneuroma in 2 cases. In the literature, approximately 15 SNF patients [14,25] underwent spinal surgery; the histopathological diagnosis was neurofibroma in all, including one who had multiple cervical and dorsal ganglioneurinomas [48]. Furthermore, 30 NF1 patients with gangliomas and NF1 have been described, but only in 5 were ganglioneurinomas located in the spine, and only 1 case had the SNF phenotype [49]. Ganglioneuromas originate from neural crest cells in the sympathetic ganglia or adrenal medulla, can be present at a young age, are sporadic or in association with NF1, and are mostly located in the posterior mediastinum, retroperitoneum or, very rarely, heterotopic areas, including sensory ganglia and nerves [50]. Our results show that not only neurofibromas, but also a rare histological type of neoplasia, ganglioneurinoma, are found in the spinal phenotype.

At present, the appearance of spinal neurofibromas seems to be an age-dependent process. The risk of developing internal neurofibromas, including spinal neurofibromas, could occur between adolescence and the age of 30 years [13]. Furthermore, we observed that some patients with spinal neurofibromas localized only in some roots developed new neurofibromas during the course of the disease and evolved to SNF. Therefore, we chose to compare our SNF patients to the patients aged >30 years without spinal tumors, because the likelihood that they could develop a spinal NF was the lowest.

Our study better defines the cutaneous features of the SNF phenotype. As already reported by Ruggieri, hyperpigmentation signs (CAL spots and freckling) were rarer in our SNF patients [11]. Conversely, our cohort showed that subcutaneous neurofibromas and, above all, internal neurofibromas were more frequently observed in SNF patients. One limitation of the research is that we did not perform a whole-body MRI scan in the cases studied; therefore, it is possible to speculate that the frequency of the internal neurofibromas reported is likely underestimated. Plotkin, who assessed the internal tumor burden using whole-body MRI, reported a statistically significant correlation between the presence of internal sheath tumors and the decreasing number of CAL spots and the presence of subcutaneous neurofibromas in NF1 patients [29]. The presence of subcutaneous neurofibromas has been associated with a higher risk of mortality [28]. Because they are not a cause of mortality, per se, they could be an indicator of a more aggressive form of the disease. The co-occurrence of spinal neurofibromas and subcutaneous neurofibromas and internal neurofibromas may suggest the presence of shared pathogenic factors. All neurofibromas arise from Schwan cells, but the Schwan cells’ interactions with axons and mostly other cells, fibroblast, endothelial cells and several components of the microenvironment, such as mast cells, macrophages, and lymphocytes, are needed for development [51,52,53,54]. A better knowledge of the tumor microenvironment and of the interactions within and between the cells that compose the different subtypes of NF is needed to understand why some body regions are particularly affected by neurofibromas.

Plexiform neurofibromas were more frequent in the SNF cases, and the difference reached statistical significance only when they were compared to the general NF cohorts reported in the literature. While in all cases studied by us, a spinal MRI scan was performed, in the previously published papers [18,24], only externally visible plexiform NF cases were counted.

The risk of developing malignancies appears to be higher in SNF cases only if they were compared to cases previously described in the NF1 cohorts from the literature [24] but not when they were compared to our classical patient cohort. All included cases were followed by highly specialized hospitals where more severe cases are referred; the difference is likely related to a biased selection.

Strong positive relationships with any type of spinal neurofibromas (i.e., single, few isolated neurofibromas, and SNF) and scoliosis were reported in a study by the Children’s Tumor Foundation, in which 2051 adult NF1 cases self-reported phenotypic traits [55]. The co-occurrence of scoliosis and spinal tumors was observed in 45% of cases by Koczkowska [9] in patients harboring missense mutations affecting NF1 codons 844-848. An association of scoliosis with the scalloping of the vertebral bodies or meningoceles but not with intraforaminal tumors were reported by Well [31], while a mild to moderate degree of scoliosis was observed only in 18% of the SNF cohort [11]. In fact, bony remodeling due the presence of tumors but also other factors, such as dural ectasias, and abnormal bony metabolism can contribute to the development of scoliosis.

Another limitation of our study is that we did not assess quality of life. Symptoms and features of NF1 are very heterogenous, some patients may experience minimal effect on their life, while others struggle with disfigurement, neurological disfunction and disability. Most previous studies assessed the quality of life in the NF1 patients in comparison with a general population, reporting a lower quality of life in the NF1 cases [56]. In order to explore whether SNF could cause a significant decrease in the quality of life, we should use suitable and specific measures such as PlexiQol [57]. Unfortunately, the test is not yet translated and validated in the Italian language, and to limit the assessment to English-speaking patients could cause an important bias. Future research should assess the quality of life and the psychosocial factors of this population.

A high heterogeneity of symptoms and features characterize NF1, and the presence of both inter- and intrafamilial variability is well known [58,59,60]. As in the SNF patients already reported, only in a minority (3 out of 19, 15.5%) of our families with all affected individuals had SNF, confirming the possible co-occurrence of SNF, MNFSR, and the classical NF1 phenotype of the same family. Both clinical features and genetic testing can help in identifying cases who are at risk of SNF and are more likely to benefit from a spinal MRI scan.

The presence of a missense mutation is associated with the occurrence of the SNF subtype, conferring an odds ratio (OR) of 3.34 (CI = 1.33–8.38); when only cases from our clinics were considered, with an OR of 4.14 (CI = 1.76–9.75); and also when SNF cases previously reported in the literature were added. Truncating and frameshift mutations, proportionally more frequent in the classical patients, lead to a loss in protein functionality, while missense mutations, observed more frequently in the SNF patients, could also lead to residual activity and a gain in the function of neurofibromin, which may have an impact on other interactors and pathways, specifically implicated in SNF, yet to be identified.

Here, we also report a higher prevalence of mutations in the SNF patients compared to the classical NF1 patients, in the C-terminal domain of the neurofibromin, containing the nuclear localization signal (NLS) and the syndecan-binding domain (SBR). The NLS domain is necessary for the nuclear localization of neurofibromin, while the function of SBR is to bind syndecans. The interaction between neurofibromin and syndecan is important for cell differentiation and proliferation, and for synaptic plasticity [61]. Functional studies are needed to confirm the possible role of C-terminal NF1 mutations in the development of the spinal form of the disease.

The mutation rate of the *NF1* gene is one of the highest known among human genes. The occurrence of germline double *NF1* gene mutations in the same subject is a very rare phenomenon when it affects both the *NF1* alleles. In fact, the mutation of both copies of the *NF1* gene is generally a lethal condition. We here report two unrelated probands with in trans double *NF1* mutations. In family 1, the proband 1136 showed a more severe phenotype with respect to the relatives carrying one of the two NF1 variants. Precisely, the mutation c.62T > A (pLeu21His), inherited from the father and shared with the brother, is a missense mutation, classified as a pathogenic variant, and it was already reported in [62]. The missense variant c.528T > A (p.Asp176Glu) was inherited by the mother, and it is reported as benign in LOVD [62,63,64]. In fact, the mother, at 55 years of age, had no tumor-related reports in her medical history, and upon physical examination, only two CAL spots, one on the left arm and one on the chest, were observed; no Lisch nodules were detected during the eye examination. An MRI scan with gadolinium showed several small (diameter of less than 1 cm) nodular-enhancing lesions in the laterocervical soft tissues, suggestive of neurofibromas, but no further sign of NF1. Thus, she did not meet the diagnostic criteria for NF1. The p.Asp176Glu substitution, predicted to be damaging by 9/20 predictors, may have a subclinical significance. Accordingly, patient 1136 showed a more severe phenotype than his affected father and brother. Similar to family 1, in family 17, the proband showed a more severe phenotype with respect to the relatives carrying one or two of the *NF1* mutations. Precisely, the mother and one of the two brothers harboring only the splicing mutation had, respectively, an MNSFR phenotype and an SNF phenotype without internal or plexiform neurofibromas. The other brother, harboring only the missense variant inherited from his father (referred to as healthy but never clinically evaluated), displayed a cutaneous NF1 form. In addition, in this case, the second variant could have a subclinical effect that could worsen the clinical phenotype of the proband carrying variants on both of the *NF1* alleles.

Moreover, in the two here-described cases, only another patient has been reported, by Fauth [65], with double in trans *NF1* mutations. The *NF1* mutational screening showed two mutations: the missense c3046T > C in exon 18 and a 3 bp deletion c8131-8133delGTT in exon 48 of the *NF1* gene. The patient showed a severe phenotype with mild dermal features, paraparesis, spinal neurofibromas, and MPNSTs at the spinal level.

These few described cases suggest that one of the two *NF1* variants maintains a residual function. In the future, expression studies should be carried out on SNF patients with two *NF1* variants, aimed at determining the possible residual activity of each of the two mutated *NF1* alleles.

## 5. Conclusions

NF1 clinical manifestations are highly variable and scarcely predictable. However, the delineation of a distinct phenotype is possible when cases are investigated using suitable diagnostic tools. Both spinal MRI and *NF1* mutational screening are useful to identify SNF cases often characterized by a few pigmentary manifestations but at high risk of problematic neurofibromas, presenting not only bilateral neurofibromas involving all spinal roots, but also a higher incidence of internal neurofibromas and nerve root swelling. Based on our results, the present clinical guidelines for the management of NF1 cases could be implemented, allowing for an earlier diagnosis, establishing an appropriate radiological surveillance, and significantly improving surgical intervention. Furthermore, new insights into the genetics of spinal NF1 could improve molecular diagnosis and counselling, opening a path for pharmacological studies focusing on the identification of new molecules targeting the pathway(s) specifically affected in this disease.

## Figures and Tables

**Figure 1 cancers-15-00059-f001:**
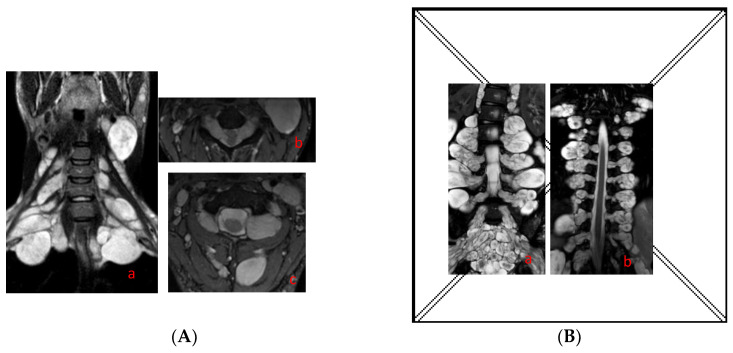
(**A**) Coronal (**a**) and axial (**b,c**) T2-weighted images showing symmetrical neurofibromas in the cervical spinal roots; (**B**) coronal (**a,b**) T2-weighted images showing symmetrical neurofibromas in the dorsal lumbar and sacral spinal roots.

**Table 1 cancers-15-00059-t001:** SNF patients’ clinical features.

NF1 Feature	Number Individuals (%) (95% CI)
**Symptomatic spinal NF**	44/81 (54) (0.43–0.64)
**Internal neurofibromas**	37/81 (45.7) (35–56)
**Nerve roots swelling**	26/81 (32.1) (23–43)
**Café au lait spots**	
<6 Cal	15/81 (18.5) (11–28)
6–10	49/81 (60.5) (50–70)
11–100	17/81 (21) (1–31)
>100	0
**Freckling**	50/81 (61.7) (50–71)
**Lisch nodules**	59/74 (79.7) (69–87)
**Cutaneous NFs**	
0	9/81 (11.1) (6–19)
1–10	36/81 (44) (34–55)
11–100	26/81 (32.1) (23–43)
>100	10/81 (12) (6–21)
**Subcutaneous NFs**	
0	23/81 (28.4) (2–39)
1–10	38/81 (65.5) (36–58)
11–100	20/81 (25) (16–35)
>100	0
**Plexiform neurofibromas**	41/81 (50.6) (40–61)
**Dural sac dysplasia**	2/81 (2.5) (0.7–8)
**Bone dysplasia**	2/81 (2.5) (0.7–8)
**Scoliosis**	34/81 (42) (32–53)
**Optical glioma**	7/81 (8.6) (4.6–16.7)
**Other gliomas**	9/81 (11.1) (5.9–20)
**Pheochromocytoma**	3/81 (3.7) (1.3–10)
**Breast cancer**	0
**MPNST**	6/81 (7.4) (3.4–15)
**Leukemia**	0
**Other tumors**	3/81 (3.7) (1.3–10)
**Neurodevelopment delay**	9/80 (11.2)(6–209
**ADHD**	2/81 (2.5) (0.7–8)
**Learning-specific disorder**	12/78 (15.3) (9–25)
**Epilepsy**	4/81 (4.9) (1.9–12)
**Headache**	8/80 (10) (5.1–18)
**Microcephaly**	0
**Macrocephaly**	19 (23.5) (15–34)
**Overgrowth**	1 (1.2) (0.2–6.6)
**Short stature**	6/81 (7.4) (3.4–15)
**Facial dysmorphism**	8/81 (9.9) (5–18)
**UBO**	37/78 (47.4) (36–58)
**Arnold Chiari**	1/81 (1.2) (0.2–6.6)
**Hydrocephalus**	6/81 (7.4) (3.4–15)
**Syringomyelia**	2/81 (2.5) (0.7–8)
**Neuropathy**	9/80 (11.2)(6–20)
**Heart malformation**	3/78 (3.8) (1.3–11)
**Vessels malformation**	10/75 (13.3) (7.4–23)
**Hypertension**	7/81 (8.6) (4.2–17)

In Bold are all NF1 features.

**Table 2 cancers-15-00059-t002:** Comparisons of clinical characteristics observed in the SNF cohort, the cohort of the “classical” NF1 phenotype from our institutions, and in previously described classical NF1 cohorts from the literature.

Title NF1 Feature	SNF (81)	Classical (106) NICB	Previously Reported NF1 Cohorts ^a^	*p*-Value SNF versus Classical NCBI	*p*-Value SNF versus Previously Reported Cohorts
Symptomatic spinal NFs	44/81 (54)	0	2/119 (1.7) 36/2058 (1.8)		<0.001 ** <0.001 **
Internal neurofibromas	37/81 (45.7)	7/106 (6.6)		<0.001 **	
Nerve roots swelling	26/81 (32.1)	3/106 (2.8)		<0.001 **	
>5 CALS	66/81 (81.5)	99/106 (93.4)	1537/1728 (89)	0. 012 *	0.013 *
Skinfold freckling	50/81 (61.7)	86/106 (81.1)	1403/1667 (84.2)	0.003 *	<0.001 **
Lisch nodules	59/74 (79.7)	87/104 (83.7)	729/1237 (58.9)	0.5	<0.001 **
Cutaneous NFs	72/81 (88.9)	99/106 (93.4)	656/723 (90.7)	0.27	0.59
Subcutaneous NFs	58/81 (71.6)	60/106 (56.6)	297/515 (57.7)	0.035	0.018 *
Plexiform neurofibromas	41/81 (50.6)	41/106 (38.7)	120/648 (18.5)	0.1	<0.001 **
Dural sac dysplasia	2 /81 (2.5)	5/103 (4.9)		0.4	
Skeletal abnormalities without scoliosis	2/81 (2.5)	6/106 (5.7)	144/948 (15.2)	0.28	0.002 **
Scoliosis	34/81 (42)	35/106 (33)	51/236 (21.6)	0.2	<0.001 **
Symptomatic OPGs	2/81 (2.5)	3/106 (2.8)	64/1650 (3.9) 7/180 (3.9)	0.87	<0.001 0.56
Asymptomatic OPG	5/81 (6.2)	5/106 (4.7)	2/45 (4.4) 70/519 (13.5)	0.66	1 0.064
Other malignant neoplasms	17/81 (21)	31/106 (29.2)	18/523 (3.4)	0.12	<0.001 **
Cognitive impairment and/or learning disability	8/80 (10)	13/101 (12.9)	190/424 (44.8)	0.71	<0.001 **
Epilepsy	4 /81(4.9)	8/106 (7.5)		0.48	
Headache	8/80 (10)	25/105 (23.8)		0.015 *	
Macrocephaly	19/81 (23.5)	31/106 (29.2)	239/704 (33.9)	0.37	0.057
Short stature	6/81 (7.4)	9/106 (8.5)	109/684 (15.9)	0.78	0.042
Facial dysmorphism	8 /81 (9.9)	14/106 (13.2)	0.48	0.65	
Neuropathy	9/80 (11.2)	5/104 (5.1)	0.1	0.13	
Cardiovascular abnormalities	13/78 (16.7)	22/97 (22.7)	54/2322 (2.3)	0.32	<0.001 *
Hypertension	7/81 (8.6)	24/106 (22.6)	0.011	0.015 *	

^a^ References [1,18,19,20,21,22,23,24,25,26,27,28,29,30,31]. Statistically significant *p*-values with an FDR of 0.05 are indicated by *, and *p*-values with an FDR of 0.01 by **.

**Table 3 cancers-15-00059-t003:** *NF1* variants observed in 19 families with at least 1 SNF case.

Family	ID Code	Subject	Phenotype	DNA Change	RNA Change	Protein Change	Type	Truncating	Exon/	Tertile	Domain
									Intron		
1	1136	Proband *	SNF	c.62T>A	r.62u>a	p.Leu21His	MS	no	2	1	nd
				c.528T>A	r.528u>a	p.Asp176Glu	MS	no	5	1	nd
	1140	Brother	SNF	c.62T>A	r.62u>a	p.Leu21His	MS	no	2	1	nd
	1139	Father	MNFSR	c.62T>A	r.62u>a	p.Leu21His	MS	no	2	1	nd
2	451	Proband	SNF	c.1393-?_2325+?del	r.(?)	p.(Ser465_Glu775de)	LD	No in-frame	13-19	1	CSRD
	494	Mother	SNF								
3	1153/176	Proband	SNF	c.6364+1G>A	r.6085_6364del280	p.Val2029Lysfs *7	SS	yes	IVS 41	3	HLR
	1202	Sister	MNFSR								
	1228	Mother	Classical								
4	258	Proband	SNF	c.2329T>A	r.2329u>a	p.Trp777Arg	MS	no	20	1	CSRD
	926	Brother	MNFSR								
5	46 B	Proband	SNF	c.5543T>A	r.(?)	p.(Leu1848 *)	NS	yes	38	3	HLR
	47 B	Mother	Classical								
6	P01	Proband	SNF	c.2297T>G	r.(?)	p.(Ile766Ser)	MS	no	19	1	CSRD
	P02	Father	MNFSR								
7	P03	Proband	SNF	c.7126+3A>T	r.7000_7126del	p.Ser2334Glyfs *21	SS	yes	IVS 47	3	HLR-CTD
	P04	Mother	SNF								
	P05	Aunt	SNF								
	P06	Cousin	MNFSR								
8	1434	Proband	SNF	c.7079dupA	r.7079dupa	p.Asp2360Lysfs *5	FS	yes	47	3	HLR-CTD
	1436	Mother	Classical								
9	1931	Proband	SNF	c.7395-?_7552+?del	r.7395_7552del18	p.Thr2466Asnfs *6	LD	yes	51	3	CTD
	1813/1912	Brother	SNF								
	1814	Mother	Classical								
10	1271	Proband	SNF	c.3827G>A	r.3827g>a	p.Arg1276Gln	MS	no	22	2	GRD
	1276	Father	MNFSR								
11	1957	Proband	SNF	c.5546G>A	r.5206_5546del31	p.Gly1737Serfs *4	SS	yes	37	2	PH
	1065	Sister	MNFSR								
	1660	Mother	Classical								
12	550	Proband	SNF	c.6791dupA	r.6791dupa	p.Tyr2264 *	FS	yes	45	3	HLR-CTD
	277	Sister	SNF								
13	1649	Proband	SNF	c.2523_2524insT	r.2523_2524insu	p.Gly842Trpfs *23	FS	yes	21	1	CSRD
	1650	Father	MNFSR								
14	1086/2277	Proband	SNF	c.6085-2A>C	r.6085_6364del20	p.Val2029Lys fs *7	SS	yes	IVS 40	3	HLR
	2198	Sister	Classical								
15	392	Proband	SNF	c.1381C>T	r.1381c>u	p.Arg461 *	NS	yes	12	1	nd
	2191	Mother	MNFSR								
16	N01	Proband	SNF	c.1527+5G>T	r.(?)	p.(?)	SS	?	IVS 13	1	nd
	N02	Sister	MNFSR								
17	N04	Proband *	SNF	c.3314+2T>C	r.spl	p.(?)	SS	yes	IVS 25	2	TBD
				c.7532C>T	r.(?)	p.(Ala2511Val)	MS	no	51	3	CTD
	N05	Mother	MNFSR	c.3314+2T>C	r.spl	p.(?)	SS	yes	IVS 25	2	TBD
	N03	Brother	SNF	c.3314+2T>C	r.spl	p.(?)	SS	yes	IVS 25	2	TBD
	N06	Brother	Classical	c.7532C>T	r.(?)	p.(Ala2511Val)	MS	no	51	3	CTD
18	N07	Proband ^#^	SNF	c.1595T>G	r.(?)	p.(Leu532Arg)	MS	no	14	1	nd
				c.3242C>G	r.(?)	p.(Ala1081Gly)	MS	no	25	2	nd
	N08	Sister	Classical	c.1595T>G	r.(?)	p.(Leu532Arg)	MS	no	14	1	nd
				c.3242C>G	r.(?)	p.(Ala1081Gly)	MS	no	25	2	nd
	N09	Nephew	Classical	c.1595T>G	r.(?)	p.(Leu532Arg)	MS	no	14	1	nd
				c.3242C>G	r.(?)	p.(Ala1081Gly)	MS	no	25	2	nd
19	N11	Proband	SNF	c.7881_7882del	r.(?)	p.(Val2627fs *)	FS	yes	57	3	CTD-SBR
	N10	Brother	MNFSR								

* Case with two *NF1* variants in trans; ^#^ case with two *NF1* variants in cis.

**Table 4 cancers-15-00059-t004:** *NF1* variants observed in 55 SNF cases.

N Patient	ID Code	Phenotype	DNA Change	RNA Change	Protein Change	Type	Truncating	Exon/	Tertile	Domain
								Intron		
**1**	**368**	**SNF**	**c.31C>T**	**r.(?)**	**p.(Gln11*)**	**NS**	**yes**	1	1	nd
**2**	367	SNF	c.58C>T	r.[58c>u, 57_60del4]	p.(Gln20Glufs*16, Gln20*)	NS/SS	yes	1	1	nd
**3**	1185/35	SNF	c.288+1137C>T	r.288_289ins288+1019_288+1136ins118	p.Gly96_Glu97ins39+fs *10	SS	yes	IVS 3	1	nd
**4**	1547	SNF	c.288+1delG	r.288delg	p.Gln97Asnfs*6	SS	yes	IVS 3	1	nd
**5**	1069	SNF	c.586+2T>G	r.480_586del107	p.Leu161Asnfs*4	SS	yes	5	1	nd
**6**	1304	SNF	c.61-?_586+?del	r.(?)	p.(Leu21Lysfs*9)	LD	yes	2-3-4-5	1	nd
**7**	1638	SNF	c.730+4A>G	r.655_730del76	p.Ala219Asnfs*37	SS	yes	IVS 7	1	nd
**8**	51 B	SNF	c.801delG	r.(?)	p.(Trp267Cysfs*14)	FS	yes	8	1	nd
**9**	741	SNF	c.945_946delGCinsAA	r.889_1062del174	p.Lys297_Lys354del	DEL-INS	no, in-frame	9	1	nd
**10**	2207	SNF	c.1246C>T	r.1246c>u	p.Arg416*	NS	yes	11	1	nd
			c.403C>T	r.(?)	p.(Arg135Trp)	MS	no	4	1	nd
**11**	425	SNF	c.1318C>T	r.1318c>u	p.Arg440*	NS	yes	12	1	nd
**12**	2171/2213	SNF	c.1711T>A	r.(?)	p.(Trp571Arg)	MS	no	15	1	CSRD
**13**	692	SNF	c.1885G>A	r.1846_1886del41	p.Gln616fs*4	SS	yes	17	1	CSRD
**14**	2146	SNF	c.2033_2034dupC	r.2033dupc	p.Ile679Aspfs*21	FS	yes	18	1	CSRD
**15**	1708	SNF	c.2252G>T	r.(?)	p.(Gly751Val)	MS	no	19	1	CSRD
**16**	1386	SNF	c.2326-3T>G	r.2326_2409del84	p.Ala776_803Glndel	SS	no, in-frame	IVS 19	1	CSRD
**17**	1493	SNF	c.2446C>T	r.2446c>u	p.Arg816*	NS	yes	21	1	CSRD
**18**	1498/61	SNF	c.2509T>C	r.2509u>c	p.Trp837Arg	MS	no	21	1	CSRD
**19**	1367	SNF	c.2810T>A	r.2810u>a	p.Leu937*	NS	yes	21	1	nd
**20**	509	SNF	c.3737_3740delTGTT	r.(?)	p.(Phe1247fs*18)	FS	yes	28	2	GRD
**21**	834	SNF	c.3827G>A	r.3827g>a	p.Arg1276Gln	MS	no	28	2	GRD
**22**	584	SNF	c.3827G>C	r.(?)	p.(Arg1276Pro)	MS	no	28	2	GRD
**23**	1145	SNF	c.3888T>G	r.3888u>g	p.Tyr1296*	NS	yes	29	2	GRD
**24**	1099	SNF	c.4267A>G	r.4267a>g	p.Lys1423Glu	MS	no	31	2	GRD
**25**	268	SNF	c.4480C>T	r.(?)	p.(Gln1494*)	NS	yes	33	2	GRD
**26**	1382	SNF	c.4719_4720dup AC	r.4719_4720dupac	p.Gln1574Thrfs*30	FS	yes	35	2	Sec14
**27**	1430	SNF	c.4773-2A>C	r.4773_5065del293	p.Phe1592Leufs*7	SS	yes	IVS 35	2	Sec14
**28**	918	SNF	c.4973_4978delTCTATA	r.4973_4978delucuaua	p.Ile1658_Tyr1659del	DEL-IF	no, in-frame	36	2	Sec14
**29**	1521/39	SNF	c.5199delT	r.5199delu	p.Ile1734Leufs*10	FS	yes	36	2	PH
**30**	1263	SNF	c.5615dupT	r.5615dupu	p.Glu1873Argfs*19	FS	yes	38	2	HLR
**31**	1803	SNF	c.5630delT	r.(?)	p.(Leu1877Tyrfs*27)	FS	yes	38	2	HLR
**32**	1450	SNF	c.5704 A>C	r.5704a>c	p.Thr1902Pro	MS	no	38	2	HLR
**33**	1242	SNF	c.5923delA	r.5923dela	p.Ile1975Tyrfs*16	FS	yes	39	3	HLR
**34**	319	SNF	c.5943G>T	r.(?)	p.(Gln1981His)	MS	yes	39	3	HLR
**35**	1478	SNF	c.5943+1G>A	r.5901_5943del43	p.Met1967Ilefs*9	SS	yes	IVS 39	3	HLR
**36**	197	SNF	c.6084G>C	r.(?)	p.(Lys2028Asn)	MS	no	40	3	HLR
**37**	334	SNF	c.6085-2A>G	r.6085_6364del280	p.Val2029Lysfs*7	SS	yes	IVS 40	3	HLR
**38**	1573	SNF	c.6085G>T	r.6085_6364del280	p.Val2029Lysfs*7	SS	yes	41	3	HLR
**39**	2281	SNF	c.6088_6090delAATinsCTTTACA	r.6088_6090delauuinscuuuaca	p.Ile2030Leufs*10	FS	yes	41	3	HLR
**40**	571	SNF	c.6311T>C	r.6311u>c	p.Leu2104Pro	MS	no	41	3	HLR
**41**	891	SNF	c.6346_6347insA	r.(?)	p.(Ser2116Tyrfs*6)	FS	yes	41	3	HLR
			c.5221G>A	r.(?)	p.(Val1741Ile)	MS	no	38	2	PH
**42**	531	SNF	c.6364+2T>A	r.spl	p.(?)	SS	yes	IVS 41	3	HLR
**43**	7	SNF	c.6688delG	r.(?)	p.(Val2230Serfs*14)	FS	yes	44	3	HLR
**44**	829	SNF	c.6791dupA	r.6791dupa	p.Tyr2264*	FS	yes	45	3	HLR-CTD
**45**	1877	SNF	c.6789_6792delTTAC	r.6789_6792deluuac	p.Tyr2264Thrfs*5	FS	yes	45	3	HLR-CTD
**46**	65	SNF	c.7846C>T	r.7846c>u	p.Arg2616*	NS	yes	54	3	CTD
**47**	NF 220	SNF	c.8051-1G>C	r.(?)	p.(?)	SS	?	IVS 55	3	?
**48**	981	SNF	c.7127-?_8314+?del	r.(?)	p.(?)	LD	no, in-frame	49-57	3	HLR-CTD
**49**	NF 291	SNF	c.-718-?_8375+?del	?	?	LD	?	/	/	/
**50**	607	SNF	/	/	/	LD-Type 1	/	/	1-2-3	/
**51**	M.E.	SNF	/	/	/	LD-Type 1	/	/	1-2-3	/
**52**	1773	SNF	NEGATIVE	/	/	/	/	/	/	/
**53**	1357	SNF	NEGATIVE	/	/	/	/	/	/	/
**54**	1390	SNF	NEGATIVE	/	/	/	/	/	/	/
**55**	1085	SNF	NEGATIVE	/	/	/	/	/	/	/

**Table 5 cancers-15-00059-t005:** *NF1* variants observed in 106 classical NF cases.

No. Patient	ID Code	Phenotype	DNA Change	RNA Change	Protein Change	Type	Truncating	Exon/ Intron	Tertile	Domain
1	623	Classical	c.200dupA	r.200dupa	p.Asn67Lysfs*10	FS	yes	2	1	nd
2	1318	Classical	c.204 + 2T > G	r.100_204del105	p.Val34_Met68	SS	no inframe	IVS 2	1	nd
3	136	Classical	c.288 + 1delG	r.288_288delg	p.Gln97Asnfs*6	SS	yes	IVS 3	1	nd
4	767	Classical	c.493delA	r.493dela	p.Thr165Leufs*13	FS	yes	5	1	nd
5	323	Classical	c.499_502delTGTT	r.499_502deluguu	p.Cys167Glnfs*10	FS	yes	5	1	nd
6	412	Classical	c.499_502delTGTT	r.499_502deluguu	p.Cys167Glnfs*10	FS	yes	5	1	nd
7	1455	Classical	c.499_502delTGTT	r.499_502deluguu	p.Cys167Glnfs*10	FS	yes	5	1	nd
8	738	Classical	c.574C > T	r.574c>u	p.Arg192*	NS	yes	5	1	nd
9	1490	Classical	c.574C > T	r.574c>u	p.Arg192*	NS	yes	5	1	nd
10	384	Classical	c.652_653delAAinsG	r.(?)	p.(Lys218Glyfs*7)	FS	yes	6	1	nd
11	858	Classical	c.653delA	r.653delA	p.Lys218Argfs*7	FS	yes	6	1	nd
12	1967	Classical	c.725delT	r.725delu	p.Met242Argfs*39	FS	yes	7	1	nd
13	1435	Classical	c.908T > C	r.908u>c	p.Leu303Pro	MS	no	9	1	nd
14	765	Classical	c.910C > T	r.910c>u	p.Arg304*	NS	yes	9	1	nd
15	329/1020	Classical	c.932_933delG	r.932_933delg	p.Gly311Glufs*6	FS	yes	9	1	nd
16	1504	Classical	c.943C > T	r.943c>u	p.Glu315*	NS	yes	9	1	nd
17	501	Classical	c.1019_1020delCT	r.1019_1020delcu	p.Ser340Cysfs*12	FS	yes	9	1	nd
18	752	Classical	c.1019_1020delCT	r.1019_1020delcu	p.Ser340Cysfs*12	FS	yes	9	1	nd
19	1542	Classical	c.1019_1020delCT	r.1019_1020delcu	p.Ser340Cysfs*12	FS	yes	9	1	nd
20	1488	Classical	c.1185+2delT	r.1063_1185del123	p.Asn355_Lys395del	SS	no inframe	IVS 10	1	nd
21	356	Classical	c.1318C > T	r.1318c>u	p.Arg440*	NS	yes	12	1	nd
22	199	Classical	c.1466A > G	r.1466_1527del62	p.Tyr489*	SS	yes	13	1	nd
23	1548	Classical	c.1466A > G	r.1466_1527del62	p.Tyr489*	SS	yes	13	1	nd
24	1669	Classical	c.1466A > G	r.1466_1527del62	p.Tyr489*	SS	yes	13	1	nd
25	1328	Classical	c.1541_1542delAG	r.1541_1542delag	p.Gln514Argfs*43	FS	yes	14	1	nd
26	915	Classical	c.1658A>G	r.1658A>G	p.His553Arg	MS	no	15	1	CSRD
27	2019	Classical	c.1907_1908delCT	r.1907_1908delcu	p.Ser636*	FS	yes	17	1	CSRD
28	1577	Classical	c.1925_ 1931 delAAATGTC	r.1925_1931delaaauguc	p.Gln642Profs*44	FS	yes	17	1	CSRD
29	663	Classical	c.2033dupC	r.2033dup	p.Ile679Aspfs*21	FS	yes	18	1	CSRD
30	1238	Classical	c.2033dupC	r.2033dup	p.Ile679Aspfs*21	FS	yes	18	1	CSRD
31	1601	Classical	c.2041C > T	r.2041c>u	p.Arg681*	NS	yes	18	1	CSRD
32	860/1782	Classical	c.2041C > T	r.2041c>u	p.Arg681*	NS	yes	18	1	CSRD
33	1754	Classical	c.2076C > A	r.(?)	p.(Tyr692*)	NS	yes	18	1	CSRD
34	524	Classical	c.2106delT	r.2106delu	p.Val703Phefs*45	FS	yes	18	1	CSRD
35	489	Classical	c.2205T > G	r.(?)	p.(Tyr735*)	NS	yes	18	1	CSRD
36	171	Classical	c.2326-1G > C	r.2252_2325del74	p.Arg752Leufs*17	SS	yes	IVS 19	1	CSRD
37	558	Classical	c.2356delC	r.2356delc	p.Gln786Lysfs*5	FS	yes	20	1	CSRD
38	507	Classical	c.2492_2493dupCA	r.2492_2493dup	p.Asp832Glnfs*10	FS	yes	21	1	CSRD
39	290	Classical	c.2540T > C	r.2540u>c	p.Leu847Pro	MS	no	21	1	CSRD
40	1590	Classical	c.2546_2546delG	r.2546_2546delg	p.Gly849Glufs*29	FS	yes	21	1	CSRD
41	53	Classical	c.2850+1G > T	r.2618_2850del	p.Lys874Phefs*4	SS	yes	IVS 21	1	CSRD
42	764	Classical	c.2851-2AT	r.2851_2990del140	p.Leu952Cysfs*22	SS	yes	IVS 21	1	nd
43	1377	Classical	c.2953C > T	r.2952_2990del39	p.Gly984_Arg997del	SS	no inframe	22	2	nd
44	1165	Classical	c.2991-2A > G	r.2991_3113del123	p.Tyr998_Arg1038del	SS	no inframe	IVS 22	2	nd
45	2022	Classical	c.2991-2A > T	r.2991_3113del123	p.Tyr998_Arg1038del	SS	no inframe	IVS 22	2	nd
46	459	Classical	c.3384_ 3390delTGGCAGG	r.(?)	p.(Gly1129Asnfs*11)	FS	yes	26	2	TBD
47	2111	Classical	c.3485delT	r.(?)	p.(Met1162Serfs*4)	FS	yes	26	2	TBD
48	1566	Classical	c.3586C > T	r.3586c>u	p.Leu1196Phe	MS	no	27	2	TBD
49	1491	Classical	c.3644T > G	r.3644u>g	p.Met1215Arg	MS	no	27	2	GRD
50	73	Classical	c.3708+1G > C	r.3497_3708del212	p.Leu1167*	SS	yes	IVS 27	2	TBD
51	876	Classical	c.3785delC	r.3785delc	p.Ser1262Leufs*4	FS	yes	28	2	GRD
52	1594	Classical	c.3826C > T	r.3826c>u	p.Arg1276*	NS	yes	28	2	GRD
53	1420	Classical	c.3870+1G > C	r.3845_3870del26	p.Lys1283fs*22	SS	yes	IVS 28	2	GRD
54	744	Classical	c.3888T > G	r.(?)	p.(Tyr1296*)	NS	yes	29	2	GRD
55	1452	Classical	c.3892C > T	r.3892c>u	p.Gln1298*	NS	yes	29	2	GRD
56	700	Classical	c.3916C > T	r.3916c>u	p.Arg1306*	NS	yes	29	2	GRD
57	809	Classical	c.3916C > T	r.3916c>u	p.Arg1306*	NS	yes	29	2	GRD
58	705	Classical	c.3941G > A	r.3941g>a	p.Trp1314*	NS	yes	29	2	GRD
59	1320	Classical	c.3975-1G > A	r.3975_3959delguuag	p.Arg1325Asnfs*16	SS	yes	IVS 29	2	GRD
60	1194	Classical	c.4077delT	r.4077delu	p.Gln1360Asnfs*25	FS	yes	30	2	GRD
61	134	Classical	c.4084C > T	r.4084c>u	p.Arg1362*	NS	yes	30	2	GRD
62	2018	Classical	c.4269+1G > C	r.4111_4269del159	p.Val1371_Lys1423del	SS	no inframe	IVS 31	2	GRD
63	1984	Classical	c.4368-1G > T	r.4368_4384del17	p.Arg1456Serfs*3	SS	yes	IVS 32	2	GRD
64	733	Classical	c.4402_4406delAGTGA	r.4402_4406delaguga	p.Ser1468Cysfs*5	FS	yes	33	2	GRD
65	965	Classical	c.4435A > G	r.4368_4435del68	p.Phe1457*	SS	yes	33	2	GRD
66	936	Classical	c.4537C > T	r.4537c>u	p.Arg1513*	NS	yes	34	2	GRD
67	1978	Classical	c.4537C > T	r.4537c>u	p.Arg1513*	NS	yes	34	2	GRD
68	170	Classical	c.4538C > T	r.(?)	p.(Arg1513*)	NS	yes	34	2	GRD
69	273	Classical	c.4630delA	r.4630dela	p.Thr1544Profs*9	FS	yes	34	2	nd
70	1353	Classical	c.4637C > G	r.4637c>g	p.Ser1546*	NS	yes	34	2	nd
71	1428	Classical	c.4854T > A	r.4854u>a	p.Tyr1618*	NS	yes	36	2	Sec14
72	1500	Classical	c.4917dupT	r.4917dupu	p.Lys1640*	FS	yes	36	2	Sec14
73	919	Classical	c.4973_4978delTCTATA	r.4973_4978delucuaua	p.Ile1658Tyr1659del	SS	no inframe	36	2	nd
74	1358	Classical	c.4981T > C	r.4981u>c	p.Cys1661Arg	MS	no	36	2	Sec14
75	32	Classical	c.5154_5157(dupATCC)	r.(?)	p.(His1720Ilefs *17)	FS	yes	36	2	PH
76	822	Classical	c.5242C > T	r.5242c>u	p.Arg1748*	NS	yes	37	2	PH
77	1749	Classical	c.5470A > T	r.5470a>u	p.Ile1824Phe	MS	no	37	2	nd
78	1213	Classical	c.5495C > G	r.5495c>g	p.Thr1832Arg	MS	no	37	2	HLR
79	966	Classical	c.5513_5514delTA	r.5513_5514del	p.Leu1838Serfs*2	FS	yes	37	2	HLR
80	213	Classical	c.5546G > A	r.5206_5546del341	p.Gly1737Serfs*4	SS	yes	37	2	PH
81	1214	Classical	c.5546G > A	r.5206_5546del341	p.Gly1737Serfs*4	SS	yes	37	2	PH (Sec14-PH)
82	502	Classical	c.5546+5G > C	r.(5206_5546del341, 5206_5749del544)	p.(Gly1737Serfs*4, Gly1737Leufs*3)	SS	yes	IVS 37	2	PH/HLR
83	474	Classical	c.5750-177A > C	r.5749_5750ins5750-174_5750-108	p.Ser1917Argfs*25	SS	yes	IVS 38	2	HLR
84	620	Classical	c.5839C > T	r.5839c>u	p.Arg1947*	NS	yes	39	3	HLR
85	946	Classical	c.5839C > T	r.5839c>u	p.Arg1947*	NS	yes	39	3	HLR
86	49	Classical	c.5890G > T	r.(?)	p.(Glu1964*)	NS	yes	39	3	HLR
87	1169	Classical	c.6084+1G > A	r.5944_6084del141	p.Ile1982_Lys2028del	SS	no inframe	IVS 40	3	HLR
88	541	Classical	c.6641+1G > A	r.6580_6641del62	p.Ala2194fs	SS	yes	IVS 43	3	HLR
89	133	Classical	c.6709C > T	r.6709c>u	p.Arg2237*	NS	yes	44	3	HLR
90	1777	Classical	c.6709C > T	r.6709c>u	p.Arg2237*	NS	yes	44	3	HLR
91	603	Classical	c.6760delC	r.6760delc	p.Glu2255Argfs*4	FS	yes	45	3	HLR
92	1327	Classical	c.6789_6792delTTAC	r.(?)	p.(Tyr2264Glnfs*5)	FS	yes	45	3	HLR-CTD
93	183	Classical	c.6789_6792delTTAC	r.6789_6792deluuac	p.Thr2264fs	FS	yes	45	3	HLR-CTD
94	29	Classical	c.6999+1G > C	r.spl	p.(?)	SS	yes	IVS 46	3	HLR-CTD
95	1467	Classical	c.7151_7161delTTGTTGCAAGA	r.7151_7161deluuguugcaaga	p.Ile2384Asnfs*13	FS	yes	48	3	HLR-CTD
96	702	Classical	c.7422dupC	r.7422dupc	p.Ser2475Leufs*6	FS	yes	50	3	CTD
97	1533	Classical	c.7486C > T	r.7486c>u	p.Arg2496*	NS	yes	50	3	CTD
98	1608	Classical	c.7500delC	r.7500delc	p.Met2501*	FS	yes	50	3	CTD
99	846	Classical	c.7926_7929delTAAG	r.7926_7929deluaag	p.Lys2643Serfs*14	FS	yes	54	3	CTD-SBR
101	727	Classical	/	/	/	LD-Type 1	/	/	1-2-3	/
102	621	Classical	/	/	/	LD-Type 1	/	/	1-2-3	/
103	/	Classical	/	/	/	LD-Type 1	/	/	1-2-3	/
104	/	Classical	/	/	/	LD-Type 1	/	/	1-2-3	/
105	/	Classical	/	/	/	LD-Type 1	/	/	1-2-3	/
106	/	Classical	/	/	/	LD-Type 1	/	/	1-2-3	/

**Table 6 cancers-15-00059-t006:** Distribution of the *NF1* pathogenic variant types between the SNF and classical groups.

Mutation Type	SNF *n* (%) (*n* = 70)	Classical *n* (%) *n* = 106	*p*-Value	OR (95% CI)	Total Number
Large deletion	7 (10)	6 (5.7)	0.28	1.85 (0.59–5.77)	13
Frameshift	17 (24.3)	38 (35.8)	0.10	0.57 (0.29–1.13)	55
Missense	15 (21.4)	8 (7.5)	0.007 *	3.34 (1.33–8.38)	28
Nonsense	10 (14.2)	28 (26.4)	0.055	0.46 (0.20–1.03)	38
Splicing	19 (27)	26 (24.5)	0.70	1.15 (0.58–2.28)	45
Deletion/insertion	2 (2.9)	0	0.16	7.7 (0.97–16.44)	2

In bold statistically significant *p*-value.* Statistically significant *p*-values with a false discovery rate of 0.05 after correction for multiple testing using the Benjamini–Hochberg procedure.

**Table 7 cancers-15-00059-t007:** Distribution of the NF1 mutation types between the SNF and classical groups, including cases reported by Ruggieri et al.

Mutation type	SNF n (%) (*n* = 70)	SNF n (%) Ruggieri et al. (*n* = 25)	Total SNF n (%) (*n* = 95)	Classical n (%) *n* = 106	*p*-Value	OR (95% CI)	Total Numbers
Large deletion	7 (10)	1 (4)	8 (8.4)	6 (5.7)	0.44	1.53 (0.51-4.59)	14
Frameshift	17 (24.3)	3 (12)	20 (21)	38 (35.8)	0.02	0.47 (0.25-0.89)	58
Missense	15 (21.4)	9 (36)	24 (25.3)	8 (7.5)	0.001 ^§^	4.14 (1.76-9.75)	32
Nonsense	10 (14.2)	3 (12)	13 (13.7)	28 (26.4)	0.025	0.44 (0.21-0.91)	41
Splicing	19 (27)	6 (24)	25 (26.3)	26 (24.5)	0.77	1.09 (0.58-2.07)	51
Deletion/insertion	2 (2.9)	3 (12)	5 (5.3)	0	0.023		5

In bold statistically significant *p*-values. ^§^ Statistically significant *p*-value with a false discovery rate of 0.05, 0.025, and 0.01 after correction for multiple testing using the Benjamini–Hochberg method.

**Table 8 cancers-15-00059-t008:** Distribution of the *NF1* pathogenic variant types (truncating/nontruncating) between the SNF and classical groups.

Mutation Type	SNF *n* (%) *n* = 67	Classical NF *n* (%) *n* = 100	*p*-Value	OR (95% CI)	Total Number
Truncating	46 (68.7)	83 (83)	0.03 *	0.44 (0.21–0.93)	129
Nontruncating	19 (28.4)	15 (15)	0.036	2.24 (1.04–4.81)	34
Nd	2 (3)	2 (2)	1	1.5 (0.2–10.97)	4

In bold statistically significant *p*-values. * Statistically significant *p*-values with a false discovery rate of 0.05 after correction for multiple testing using the Benjamini–Hochberg procedure.

**Table 9 cancers-15-00059-t009:** Distribution of the *NF1* pathogenic variants in different *NF1* gene regions between the SNF and classical groups.

Region	SNF *n* (%) *n* = 67	Classical NF *n* (%) *n* = 100	*p*-Value	OR (95% CI)	Total Number
CSRD	11 (16.4)	16 (16)	0.94	1.03 (0.44–2.38)	27
TBD	1 (1.5)	4 (4)	0.65	0.36 (0.4–3.33)	5
GRD	7 (10.4)	19 (19)	0.13	0.49 (0.2–1.26)	26
Sec14-PH	5 (7.5)	8 (8)	0.89	1.04 (0.34–3.15)	13
HLR	17 (25.4)	12 (12)	0.025	2.5 (1.1–5.64)	29
HLR-CTD	6 (9)	4 (4)	0.2	2.36 (0.64–8.7)	10
CTD-SBR	1 (1.5)	1 (1)	1	1.5 (0.09–24.4)	2
CTD	2 (3)	3 (3)	1	0.99 (0.16–6.11)	5
NLS	0HLR	0	-	-	0
SBR	0	0	-		0
Others	17 (25.4)	33 (33)	0.29	0.69 (0.34–1.37)	50

n = number of *NF1* mutations from the analyses; whole gene deletions were excluded. Some mutation locations may overlap different regions. In bold statistically significant *p*-value. *p*-values of 0.025 after correction for multiple testing using the Benjamini–Hochberg procedure with a false discovery rate of 0.25 did not remain significant.

## Data Availability

Raw reads of the NGS data are available in the NCBI Short-Read Archive (SRA, https://www.ncbi.nlm.nih.gov/sra/) (accessed on 18 July 2022) under accession number: PRJNA8509016.

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
