# Peer review of "A Translational Approach to Spinal Neurofibromatosis: Clinical and Molecular Insights from a Wide Italian Cohort"

_cancers, 2022, doi:10.3390/cancers15010059_

Round 1

Reviewer 1 Report

Summary

The authors present a series of 81 patients with SNF discovered through an electronic database (IRCSS), describing genetic, phenotypic and familial variability of the patient cohort.

Strengths

The study is relevant to the field of Cancer research. The manuscript is overall well written.

Weaknesses

1.       Minor grammatical errors are present

a.       In the second paragraph of the Introduction section, the phrase, “This feature allows to specifically distinguish SNF…” should be rephrased to, “This feature allows one to specifically distinguish SNF…”

b.       In the last paragraph of the Introduction section, the phrase, “…features characteristic for sporadic and familiar cases with SNF…” should be rephrased to, “…features characteristic of sporadic and familial cases with SNF…”. In fact, it seems the term “familiar” is used throughout the manuscript when “familial” may be better suited.

c.       In the first sentence of the Discussion section, “To our knowledge this is still the largest series of SNF patients reported, until now, all cases…” should be edited to read, “To our knowledge this is still the largest series of SNF patients reported, until now. All cases…”

d.       Numerous other examples are present and won’t be exhaustively reported

Reviewer 2 Report

The authors present an interesting analysis of 81 patients with SNF, 55 from unrelated 37 families and 26 belonging to 19 families with at least one member affected by SNF and 106 NF1 patients aged > 30 without spinal tumors. A comprehensive NF1 mutation screening was performed using a NGS panels including NF1 and several RAS pathway genes. The main features of SNF subjects were a higher number of internal neurofibromas (p<0.001), nerve-root swelling (p<0.001), subcutaneous neurofibromas (p= 0.03), while hyperpigmentation signs were significantly less frequent compared with classical NF1-affected cohorts (p=0.012). Also a phenoypic variability within SNF families was observed. The proportion of missense mutations was higher in SNF cases than in the classical NF1 group .

15 patients underwent neurosurgical intervention. The histological findings were neurofibromas in 13 and ganglioneuromas in 2.

The authors concluded that their large series allows a better definition of clinical and genetic features of SNF patients. This could improve the management and genetic counseling of NF1.

The manuscript is well written and the scientific methods are good. The results are well presented. However, is suggest to comment to following aspects
before the manuscript might be considered for publication

1. the authors reported that only 18.5 % underwent surgery. Could the authors analyze the impact of surgery of the further clinical course and follow up of these large cohort. At what time and at which tumors, there is a necessity for surgery?
2. the authors should also add the clinical data of follow up: PFS, complaints and clinical setting of these patients. QoL of NF1 and SNF patients. This is also important for the better understanding of the disease.
